# Giants in the Frame: A 1964 Photo Analysis of How Malcolm X and Dr. Harry Edwards Connected Race, Religion, and Sport

**Whitney Griffin** [1,*] and **C. Keith Harrison** [2,*]

1   Department of Psychology, Cerritos College, Norwalk, CA 90650, USA
2   College of Business Administration, University of Central Florida, Orlando, FL 32816, USA
*   Correspondence: wgriffin@cerritos.edu (W.G.); carlton.harrison@ucf.edu (C.K.H.)

**Abstract:** Racial analysis of photography in the canon is important when unpacking layers of racial discrimination, Black manhood, and the historical dynamics within social forces that create stereotypical perceptions of African American males. Race, religion, and sport allow scholars to unpack the perceived disposability of Black lives in contemporary society. In an effort to fully understand how sport and religion inform racialized experiences in Black manhood, the current paper seeks to advance theories of visual and racial culture in a particular context. Contextual analysis of a 1964 photograph of Malcolm X and Dr. Harry Edwards synthesizes the visual turn and offers insight into how race, sport, and religion collide to raise minority pride. A contextual analysis accounts for the ways in which visual materials function within broad social ecologies of Black masculinity. Implications are discussed for the role of sport and religion in continuing activism for racial equality.

**Keywords:** photo analysis; Black church; racism; sport activism; visual theory

## 1. America Is Built on Movements

Photographs are a timeless stamp of moments in human existence. The world of sports is ripe with robust imagery, visual symbolism, and psychological interpretation that both reflect and generate the sociocultural context in which they were produced. In sport history, the role of visual arts "is to challenge and change perceptions, to encourage the viewer to see familiar things differently" (Huggins 2008, p. 312). Racial analysis of photography in the canon is important when unpacking layers of racial discrimination, Black manhood, and the historical dynamics within social forces that create stereotypical perceptions of African American males. The domains of race, photographs, and sport are interdisciplinary in nature and their overlap encourages the viewer to see familiar art differently.

Race and religion are shaped by social practices and economic conditions. Both involve bodily practices, displays of power, and complex histories of mass injustice (Lloyd 2013). Visual arts within race, religion, and sport allow scholars to unpack the perceived disposability of Black lives. Professional Black male athletes in particular hold powerful social positions that influence people's notions about performances of gender (Rhoden 2006). Since images of social movement leaders and athletes are central to how Black men conceptualize manhood and deconstruct harmful stereotypes about masculinity (Goodwill et al. 2019), it is worthwhile to investigate photographs of such cultural figures. The interconnection of religion and racial inequality is increasingly necessary; there is an "ongoing urgency of understanding race and religion as two key features of American life that shape the distribution of resources, life chances, and domination and oppression" (Husain 2017, p. 2). It is urgent and incumbent upon scholars to design sociological frameworks that account for the ways in which religion operates within racialized experiences. One scholar notes, "Given the long tradition of Christianity in the Black community, it is possible that Black Christians today are navigating society in a way that existing literature and frames

do not equip sociologists to fully understand" (Allen 2019, p. 8). Existing scholarship on key features of American life must expand.

In an effort to fully understand how sport and religion inform racialized experiences in Black manhood, the current authors seek to analyze, synthesize, and offer insight into how race and religion collide to illuminate the advancement of Black masculinity as it is manifested in sports. Within the saturated context of America in the 1960s, a surreptitious photo captured two Black men who had a ripple effect in the ongoing struggle to end racial discrimination. Individually, Malcolm X and Dr. Harry Edwards promoted Black liberation and nationalism. Together, they directly influenced sport history. An analysis of their image representation in sport history provides scholars with an opportunity to synthesize those theories that drive a *unique* and *cultural* interpretation. Social movements gain momentum over time and as Dr. Edwards said, "America is built from movements" (Kamoji 2021). Using this critical approach to visual culture, we generate new insight into how one social movement gained momentum when race, religion, and sport link manhood to the collective struggle.

## 2. Review of Literature

### 2.1. The Culture and Context of 1964

America's climate in the 1960s was a turbulent time for the definition of freedom. Multiple social movements sprang concurrently to increase democracy and end inequality. These movements included the anti-Vietnam War movement, the women's movement, the gay rights movement, and the student movements precipitated by Black college students who formed the Student Nonviolent Coordination Committee (SNCC). Nonviolent images of political protests circulated the world and brought attention to the Civil Rights Movement, e.g., Rosa Parks sitting on a bus in 1955; Dr. Martin Luther King Jr.'s "I have a dream" speech in the 1963 march on Washington; Black students attending a recently integrated school in 1963 Birmingham, Alabama; Tommie Smith and John Carlos displaying the Black power symbol during the 1968 Olympic medal ceremony.

Peaceful protests were not the only response to centuries of slavery and socioeconomic discrimination. In contrast, the late 1960s saw more than 750 riots across the urban United States (Blackman 2021). During this destructive time at home and abroad, American political leaders sought to project an image and narrative in the Cold War that systemic racism was no longer a reality. The integration of Black athletes into sports was used as propaganda for a post-racial, cohesive American society (Blackman 2021). Black Christians double-qualified as the ideal content for religious, nationalist propaganda at a time when America faced the threat of communism (Putz 2022). Black athlete success helped project the image of racial inclusion and boosted American nationalism. In comparison to the racial superiority and eugenics campaigns of Nazi Germany, American sport integration seemed to offer potential social change when Jesse Owens went to Germany and won four Olympic gold medals and heavyweight boxing champion Joe Louis beat his German contender with a first-round knockout in 1938 (Putz 2022). However, the Black Power salute in front of a global audience in 1968 signified rebellion against White supremacy: "for the first time in U.S. history, the reigning super-power had its own human rights drama unfold on a world stage" (Smith 2016, p. 70). Unforgettably, sport was a platform to inform the world that America was not full of the freedom it claimed.

Since the first enslaved Africans were brought to what would become the United States, the Black body has been framed in a biased, deficit model. The 1965 Moynihan Report employed a deficit framework to rationalize why Blacks were psychologically incompetent and why Black men were especially irresponsible fathers (Givens et al. 2016; Hunter and Davis 1994). Yet, promoters of anti-Black racism have used scientific racism, the combination of pseudoscience and flawed studies, to justify racial inequality and White supremacy long before the 1960s. In the 19th century, scientists used biology-based racism, e.g., craniometry and phrenology, to claim that White people were biologically superior to Blacks. The innate biological inferiority mythology is the racialized physical superiority

that ignores and insults the humanity of Black people: "The black body had always received attention within the framework of white supremacy, as racist/sexist iconography had been deployed to perpetuate notions of innate biological inferiority." (hooks 1995, p. 203). In the 20th century, Black male athletes began to destroy these insidious stereotypes on a national level, e.g., Jack Johnson as the first Black heavyweight boxing champion in 1910, and international level, e.g., Arthur Ashe as the first Black male to win the Australian Open in tennis in 1968.

In response, some sport scientists and sociologists have legitimized newer stereotypes about Black athlete superiority in conjunction with White intellectual superiority (Dyreson 2008; Hall 2002; Sailes 1991, 1993). At either end of the performance spectrum, Black athletes have been placed in a deficit in comparison to Whites. In 1971, Edwards wrote "The Sources of the Black Athlete's Superiority" shortly after the 1968 Olympics. In this article, he cogently connected the myth of Black biological superiority to the attention that Black athletes received from winning so many games:

> " . . . in recent years the former [the Black male athlete's inherent physical superiority] has been subject to increasing emphasis due to the overwhelmingly disproportionate representation of black athletes on all-star rosters, on Olympic teams, in the various 'most valuable player' categories, and due to the black athletes' overall domination of the highly publicized or so-called 'major sports'—basketball football, baseball, track and field." (Edwards 1971, p. 34).

A decade later, in one of his most notable contributions to sport sociology research, Dr. Harry Edwards coined this iteration of scientific racism the dumb Black jock myth (Edwards 1984). Sport sociologists have continued to dismantle unsubstantiated claims of racially determined athletic superiority ever since.

### 2.2. The Black Church and Religious Commitments to Black Advancement

As the oldest social institution in the Black community, the Black church has always been a significant place of human connection and affirmation. From their earliest history as Americans, Black people were enslaved, silenced, and dehumanized. In the 18th century, rudimentary praise houses were built by and for slaves. In his book, *The Black Church: This is Our Story, This is Our Song*, Gates (2021) depicts how these meeting grounds served as an informational marketplace since slaves were not allowed to socialize in public. Although the inhumane politics of U.S. chattel slavery prohibited the structure of an organized Black family, slaves were still pushed to convert to Christianity. Thus, the Black church became the oldest, first economically self-sustaining institution fully controlled by Black people (Billingsley and Caldwell 1991; Wortham 2009). According to Gates, the Black church was the bedrock for Black culture and political action. As the institution survived slavery, it was the context for the first sociological study on religion in W. E. B. Du Bois's 1903 text, *The Negro Church* (Allen 2019).

Within a system of racial inequality and oppression, religious spaces offered Black people the opportunity to develop racialized attitudes, spiritual liberation, and oppositional consciousness. This opportunity for meaning-making was essential for the Black freedom struggle: "More specifically, Christianity provided a means to affirm their humanity while existing within a racial hierarchy that otherwise deemed them less than fully human" (Allen 2019, p. 2). From slavery through emancipation to the current hostile racial climate in America, the Black church has provided a multiplicity of functions: physical and psychological social refuge (Gilkes 1998; Morris 1984; Pattillo-McCoy 1998; Peck 1982), schoolhouse (Morris 1996), and a center for leadership and political mobilization such as the Civil Rights Movement (Allen 2019; Billingsley and Morrison-Rodriguez 1998; Calhoun-Brown 2000; Sorett 2017; Smith 1996). Overall, the Black church is and has always been an incubator for Black resistance to White-supremacist capitalist patriarchy.

Among its many functions as a community-based organization, the Black church continues to serve as an ideal place for social belonging and personal identity in a culture that problematizes Black men. Recent research on identity development illustrates the

ways in which religious organizations provide an exploratory space for resisting racist stereotypes and cultivating positive aspects of Black identity and self-worth (Hickman-Maynard 2018; Kyere and Boddie 2021; Martin et al. 2021; Stanczak 2006). The Black church is especially helpful for youth, as researchers have found that racial identities and a community connection are both linked to better academic performance (Altschul et al. 2008; Byrd and Chavous 2009; Oyserman et al. 2003). The pedagogy of care inherent in the Black church also facilitates the development of racial identities through the role of spiritual advisors and mentors. Caring relationships between youth and adults are an important aspect of identity-building work for young Black men because they can access identity options that expand beyond the limited stereotypes of masculinity presented to them. A caring environment in community-based organizations facilitates agency for Black male role modeling, emotional management, intimate connection to others, and a contribution to society (Givens et al. 2016). The overlapping theme of liberation in Christian theology and secular Black Nationalism both allow agentic challenges to racism.

The Black church and sport history are intersecting spheres of racial pride and strength. Anthropologists have traced the history of sport and religion throughout the last 3000 years: from primitive ritual ball games in exchange for fertility from the gods to ancient Greek Olympic games to pay homage to gods and goddesses, to the philosophical movement of Victorian muscular Christianity (Parker and Watson 2013). Religion has historically manifested in sport through public prayer and displays of heroism (Hoffman 2010). Scholars have paid particular attention to contemporary Black athletes and their relationships with religion, such as Kobe Bryant and the influence of Catholic teachings (Mazurkiewicz 2021), Tiger Woods and Buddhism (Thangaraj 2020), and Kareem Abdul-Jabbar's path to self-actualization through his conversion to Islam (Goudsouzian 2017). Jones (1991) posited that both church and sports appeal to Black Americans because of their overlapping structures of ritualized order, structure, and discipline: "These certainties are reassuring to African Americans and a source of strength" (p. 2). In particular, street basketball has been interpreted as a "lived religion" for inner-city young urban Blacks engaging in ritual practices on the court (Woodbine 2016, p. 9). Empirical findings further suggest that athletes and civil rights leaders may play an influential role in how Black men construct their masculine identities (Goodwill et al. 2019). In both sport and religious organizations, Blacks have demonstrated strength, mastery, and control.

### 2.3. Black Masculinity/Manhood in America

Being Black and male in American society places one at risk for unemployment (U.S. Bureau of Labor Statistics 2020), school dropout (National Center for Education Statistics 2018), and state or federal prison (Bureau of Justice Statistics 2018). The events before and after the killing of Trayvon Martin, Mike Brown, and George Floyd necessitate a reimagining of the criminal, anti-intellectual stereotypical identities ascribed to Black manhood. In a patriarchal White supremacist society with historical and perpetual oppression, masculinity holds a unique and paradoxical place in the discussion of the Black male condition: "Undereducated and unemployed black men are the raw material that society pays its police to contain or sweep conveniently from view" (Guerrero 1995, p. 396). The sociocultural context of male role identity for Black men includes expectations to conform to both traditional gender requirements, e.g., economically successful, competitive, aggressive, virile, physically strong, and ethnic requirements, e.g., cooperation, promotion of Black community (Franklin 1999). Negotiating manhood when these role identities conflict promotes diverse and complex notions of masculinity. Although early studies on manhood only included White masculinities (Goodwill et al. 2019), recent scholarship has revealed that Black men begin navigating challenges to their masculinity and stereotypical identities early in middle school (Givens et al. 2016; Nasir 2011; Nasir and Shah 2012).

In an anti-Black society, mass media is largely responsible for circulating images that erase and silence Black perspectives and vulnerabilities (Rudrow 2019). Cooper (2006) posited that the reason that such a bipolar visual representation of Black masculinity is to



help resolve White mainstream's anxiety post-civil rights. Such schizophrenic perceptions and assumptions (Guerrero 1995) justify the simultaneous exclusion of most Black men, i.e., jail or low socioeconomic status, with the inclusion of a few token, "white-acting Black men into the mainstream" (Cooper 2006, p. 854). In between these binary options, "it is imperative that as scholars and readers of gender we move beyond and against the 'bipolar' black masculine representation" (McCune 2012, p. 123). Standing in contrast to the privileges of White masculinity, from a young age, Black boys are often taught how to regulate their physicality through body gestures, vocal expressiveness, and cool stoicism (Majors and Billson 1992; Rudrow 2019; Stoever 2016). Resistance or failure to comply with these cultural scripts can lead to harm and/or death.

Members of dominant groups maintain their social privileges by rendering marginalized groups invisible and/or hypervisible (Settles et al. 2019). It is important to note that the complexities of visibility, hypervisibility, and invisibility are not mutually exclusive within Black masculinity. Rather, a Black man could be visible in one context and hypervisible in another, i.e., elite athletes in football and men's basketball. Marginalized groups respond by managing their visibility, many times by downplaying or hiding parts of themselves. While managing visibility, stigmatized minority groups are likely to experience additional rejection, prejudice, and discrimination due to their group membership and token status (Settles et al. 2019). Chronic, negative psychological and physiological health consequences (especially cardiovascular outcomes) of racism and prejudice have been documented for people of color in the social sciences and public health literature (Clark et al. 1999; Giscombé et al. 2005; Harrell 2000; Richman et al. 2007; Sawyer et al. 2012; Sellers et al. 2006; Solórzano et al. 2000). Rather than trade their sense of authenticity and social belonging (Settles et al. 2019), it is understandable that Black men would look to religious organizations for recognition in ways that affirm their identities.

A new conception of manhood arose during the Black Power era. As Black nationalists opposed racial oppression and police brutality, Black Panther Party activists responded with armed resistance. In contrast to the nonviolent strategies of the Civil Rights Movement, the Black Power movement presented a counternarrative to the traditional stereotypes of the powerless Black male: "Among Black Power militants and their black nationalist precursors, by contrast, self-defence, while initially intended to stop police brutality and other forms of racist oppression, ultimately came to be utilised mainly as a symbol of militant black manhood" (Wendt 2007, p. 544). Such symbols disrupted traditional gendered stereotypes for men and replaced powerlessness with a positive Black identity.

The Nation of Islam (NOI) was a religious group that merged the belief systems of Islam with Black nationalism. The NOI emphasized Black liberation, self-discipline, self-sufficiency, and rejection of racial integration. This philosophy reinforced the message of Black Power, which taught Black Americans to have economic independence in their businesses, schools, and community organizations (National Museum of African American History and Culture n.d.). Symbolically, the boldness of armed resistance affirmed Black masculinity in a more defiant way than the Civil Rights Movement did. This rejection of nonviolence tactics was embraced by members of the NOI and their spokesman, Malcolm X:

> "From Malcolm's perspective, armed resistance represented a crucial affirmation of black manhood. Rather than follow [Martin Luther] King's emasculating philosophy, he argued, black men needed to regain their role as protectors of 'their' women and their families." (Wendt 2007, p. 554).

The role of Islam in Malcolm X's relationship with Black masculinity was largely based on racial uplift and defiance of racist, powerless stereotypes. Though rooted in traditional gender roles that perpetuated the subordination of Black women, the militant masculinity of the NOI affirmed male members as the guardians of their female counterparts. Malcolm's ideas on defensive violence, active armed resistance, and manhood had a direct impact on Huey P. Newton and Bobby Seale, the founders of the Black Panther Party: "Like their hero Malcolm X, Newton and Seale rejected the non-violent protest as degrading to black

masculinity, offering an alternative construction of manhood that was grounded primarily in the use of violence to defend the black community" (Wendt 2007, p. 556). This identity formation presented Black men as warriors who were worthy of respect and capable of protecting their families.

The new conception of manhood combined with community activism for working-class families made the Black Panther Party an ideal organization for young Black male athletes. Thus, Dr. Edwards sought the endorsement of the Black Panther Party as he began to organize the revolt of the Black athlete through his Olympic Project for Human Rights (OPHR):

> "So, we tried to cultivate the black youth movement. And when it comes down to black athletes, you're talking about that level, that spectrum of black society that really the Black Panther Party was attempting to organize and coordinate. The churches did not really focus on that stratum of black youth. The Civil Rights Movement most certainly did not focus on that stratum of black youth. The two institutions that really cultivated that stratum of black youth were the military and the coaches, the athletic sector, because that's where the athletes come from." (Wilmot 2005, pp. 69–70)

The Black athletes that Dr. Edwards influenced most notably were Tommie Smith and John Carlos. Their protest during the 1968 Olympic Games in Mexico City was in coordination with Edwards and their vision for racial uplift. Malcolm X influenced Black athletes as well, most notably Muhammad Ali and Kareem Abdul-Jabbar's conversion to Islam (Goudsouzian 2017; McElvain 2022).

*2.4. Harry Edwards and Malcolm X*

By the time he wrote his dissertation, Harry Edwards was already a published scholar. He had expanded the field of sport sociology through two books and over 25 articles on Black students and the revolt of the Black athlete. Most notable to the current discussion, however, is his 1968 article on the dynamics between Muslim and Christian religions in the Black community entitled "Black Muslim and Negro Christian Family Relationship." Through a series of interviews, participant observations, and informants, Edwards compared family life between those affiliated with the Nation of Islam (NOI) and those with lower-class Black Christian churches. In comparing the different patterns of family life between the two groups, e.g., husband–wife relationships, family–extended kin relationships, and family–community relationships, he revealed that the Muslim participants were more aligned with the "wage-earning, noncriminal, middle-class ideal" than the lower-class Black Christians (Edwards 1968, p. 610). Thus, while the mass media portrayed Muslims as the rebellious "other" who refused social conformity, the results of his study portrayed them as meeting traditional, American middle-class goals more than Black Christians.

In the context of the Civil Rights Era and the racist dynamics of America, the crusade for change was galvanized by religious organizations. While Dr. Edwards was studying the connections between Black Christians and the NOI, Malcolm X had quickly become one of the NOI's most influential leaders. Born Malcolm Little, Malcolm encountered the teachings of Elijah Muhammad while in prison for larceny. Soon after, he adopted the last name "X" to reject his slave name and represent his unknown African ancestral name. As a follower of Elijah Muhammad and his group of Black Muslims, Malcolm X became the minister of a mosque in Harlem, New York, where his electric oratory skills helped expand the NOI from 400 members in 1952 to 40,000 by 1960 (History.com 2022). In 1963 he split from the NOI and in 1964 traveled to Mecca, Saudi Arabia, where he experienced a spiritual transformation. He returned to America with the name El-Hajj Malik El-Shabazz and a new perspective on activism. Unlike the NOI religious group that identified White people as the devil, his spiritual journey to Mecca influenced him to abandon his belief in violence and self-defense against White aggression.

The deep commitments of Dr. Edwards and Malcolm X to defending human rights and raising minority pride led each man to create Black nationalist organizations. After

his return from Mecca in 1964, Malcolm X founded the Organization of African American Unity (OAAU), a pan-Africanist organization dedicated to fighting for Black American human rights and promoting cooperation among all people of African descent in America. Three years later, after their documented encounter, Dr. Edwards founded the Olympic Project for Human Rights while a professor at San Jose State, where John Carlos and Tommie Smith were students. Dr. Edwards established the OPHR to protest racial segregation in America and South Africa, and fight racism in sports in general.

Serendipitously, Edwards met Malcolm X in 1964 after his separation from the NOI and his spiritual transformation. Then a doctoral student on a fellowship at Cornell University, Edwards was conducting his research at a Black church. His thesis focused on Black people's perceptions of role models, including Black sports heroes. Thus, the sociological pursuits of soon-to-be Dr. Edwards unified religion, race, and sport. This collision took him to Harlem, where he was near Malcolm X, and someone captured a photo of the two men. While it is not documented if Malcolm participated in Edwards' dissertation research, it is important to note that these men met after Malcolm's spiritual transformation. His journey to Mecca changed his perspective on Black activism in America so much that he swiftly created a pan-African organization to combat White supremacy upon his return. Before his spiritual journey, his separatist views would have made for a different dialogue between the two men, if one indeed happened during their encounter.

*2.5. Image Representations of Black Men in American Culture*

Image representations of Black men in America are controlled by mainstream Hollywood's formulaic narrative. This hegemonic account in popular culture includes a racist hidden political agenda, carefully crafted images of racial identity, a closed system of nepotism, and executives who are free to subscribe to their own politics (Bourne 1990; Miller 1998; Rhodes 1993). Beyond the exclusion of independent filmmakers and the control of images in American films, the Hollywood system ensures public exposure of their images, i.e., by monopolizing marketing and distribution, and manufactures the audience demand for the films they offer (Bourne 1990; Miller 1998). The effectiveness of the hidden, covert racist activity is that it operates "in all aspects of commercial American cinema and, thus, defines how blacks are portrayed on the screen which, in turn, defines how black audiences see themselves" (Miller 1998, p. 19). The definitions of Black male identity are representations that play into the White imagination, which is a limited one.

Hollywood's racist political agenda positions Black masculinity at extreme ends of representations. The polarity of good and bad images ignores the nuances and realities of Black experiences. Good Black men are scripted as shallow, sterile, and servile, e.g., many of Sidney Poitier's characters reassured White people of their innocence and superiority (Guerrero 1995). On the other extreme end, the first person to ever shoot a film in Hollywood portrayed Black men as blatantly evil in order to convey the same reassurance of White superiority, i.e., D. W. Griffith's sexually threatening and primitive characters in *Birth of a Nation* (1915). Henry (2002) has suggested a correlation between the way Black masculinity is enacted on film and how it is defined in the White imagination. Portrayed as a threat to middle-class notions of White femininity and family, these representations of Black manhood are violent, incompetent, predatory, prone to criminality, and hypersexual (Collins 2004; Ferguson 2000; Gray 1995; hooks 2003). These character traits have been grouped into a few racist, common, reductive archetypes created in an effort to control Black men's bodies the during post-emancipation Jim Crow era where the Black rapist myth was perpetuated (Rudrow 2019). Among the archetypes to emerge from antebellum America was the Uncle Tom, an acquiescent, faithful, and happy servant (Bogle 1994), and the buck, a devious, lascivious, and mindless brute (Jackson 2006). Stereotypes of Black masculinity are central to the image constructions that stem from the White imagination: "At the center of the way Black male selfhood is constructed in white-supremacist capitalist patriarchy is the image of the brute, untamed, uncivilized, unthinking and unfeeling" (hooks 2003, p. xii). One of the many contributions of the Black Power movement in the

1960s was radicalizing the Uncle Tom imagery of the "emasculated and shuffling Black male dictated by racial caste" (Hunter and Davis 1994, p. 23). This was an important shift in raising minority pride and imagery of Black nationalism.

Negative images of Black masculinity performance are harmful when they distract from issues of systemic inequality. Media representations of Black men are used by the dominant White class to displace attention away from issues like economic inflation and racism, instead inciting panics about crime and middle-class security (Gray 1995). These threats of excessive Black masculinity include hyper-aggressiveness, lack of accountability, hypersexual, and appearance of wealth (Hunter and Davis 1994; Majors and Billson 1992). Pairing Black manhood with "non-achiever or criminal in the social imagination of our society" is especially destructive for Black male youth who shape their racial identities by drawing upon stereotypes (Givens et al. 2016, p. 170; Way et al. 2013). Images that evoke fear and dread send messages that Black men are less capable of humanity: "All that many young black men have to model themselves after is the media's definition of who they are, and a cycle of destruction of the black family is allowed to propagate" (Miller 1998, p. 22). However, image representations of Black men are evolving in film and research, most notably in films written and directed by Black men. Boylorn (2017) illustrated the sophisticated code-switching of Ryan Coogler's main character in *Fruitvale Station* (2013), which allowed a nuanced performance of Black masculinity that is more complex than the unidimensional images available in mainstream visual culture. Furthermore, empirical data add to the existing knowledge on Black masculinity by documenting how Black men can navigate both their sexual and masculine identities without hypersexualizing their conception of manhood, especially when referring to well-known social movement figures and athletes (Goodwill et al. 2019).

The historical images of Black men as hypersexual, aggressive criminals inform how contemporary Black male athletes are represented through the White gaze. The voyeuristic fear and fascination with Black male bodies have traditionally been managed by White America through minstrelsy and lynching (Tucker 2003). Ferber (2007) even goes as far as naming it an "obsession with controlling and 'taming' Black male bodies" (p. 11). Professional basketball in particular has been a site of racialized containment that creates hypervisibility around disciplining Black men. For example, Dennis Rodman blatantly disregarded National Basketball Association (NBA) regulations and provided a counternarrative to Black male subservience:

> "His refusal to obey even the most basic rules for NBA team members, combined with his occasionally angry outbursts toward people in and around the court during games, fanned the flames of public fears about the sorts of rules for which he might harbor similar disregard in off-court settings. Ironically, the rules of which he was most frequently in violation in off-court settings had no relationship to actual acts of violence or criminality of any kind." (Tucker 2003, p. 320)

His flamboyant and controversial behavior pushed the boundaries of the politically neutral Black man that White America identified for Black athletes. Rodman was perceived as threatening because he was not easily controlled. Disproportionate media coverage that demonizes Black male athletes as "'bad boys' is used as a tool to exert control over those men who do not so easily submit to White male authority" (Ferber 2007, p. 20). These representations are evident as early as high school. In a Toronto Star newspaper, the *High School Report*, Canadian athletes were framed in a racialized manner. White student athletes were constructed as having a "nifty 93 percent academic average" and recognized for their academic achievements (Saul and James 2006, p. 72). On the other hand, the disciplinary problems of Black athletes were overemphasized, highlighting the top Black high school runner's verbal assault and expulsion from school.

Although White athletes also face violent and sexual assault charges, Black male athletes receive hypervisible media coverage that reifies the stereotype of Black men as inherently dangerous and fearful. Media commentators tend to use language that characterizes White athletes as hard-working, emotionally mature, and intellectually superior

(Hoberman 1997). In contrast, Black athletes are portrayed as biologically talented, emotionally immature, physically aggressive, hypersexual, and intellectually inferior (Hawkins 2010; Hughey and Goss 2015; Sailes 1991; Tucker 2003). A recent content analysis of Black and White athletes featured in *Details, Esquire, GQ,* and *Playboy* revealed that while Black athletes were observed on more magazine features, White athletes were represented in 2.5 times as many sports (Denham 2020). Although basketball stars such as LeBron James and Kobe Bryant were celebrated, the magazines restricted the Black athlete experience by showcasing more than 80% of Black athletes in team sports than individual, i.e., basketball, football, and boxing. Young Black male athletes are thus exposed to a limited number of sports to develop their physical talents, further perpetuating the myth of White superiority.

*2.6. Intersection of Visual Theories*

Photography of sport representations matters because scholars can maximize the medium to archive, divulge, and appraise social issues. Since the beginning of the 21st century, an explosion of visual culture analysis in the humanities and social sciences has marked a "visual turn" that has rippled into the landscape of sports history (Huggins 2008). The visual turn is applicable to the convergence of sport, religion, and race because it signifies an engagement with both the content and the form of a visual representation. According to O'Mahony (2018), such an analysis needs to consider the visual conventions, techniques, devices, medium of the image production, and how these factors combine to shape complex readings of the material. Beyond the content and production, researchers should also consider the potential reception of an image within both spatial and temporal contexts. Thus, America's social climate in the 1960s is foundational to any visual analysis of the photo presented. The development of the visual turn as a method for engaging with visual culture (Burke 2001; Mitchell 1994; Rose 2007) allows for a critical approach to the materials of sport history. The purpose of the visual turn within sport history provides the opportunity to

> "facilitate a more critical and analytical approach to the deployment of such material, enabling a more nuanced engagement with the visual and material culture of sport as a rich and valuable resource both to facilitate and to problematize explorations into sport's many pasts." (O'Mahony 2018, p. 29)

In the age of information, people can engage with sport using visual options from television, still photography, video games, and other digital media. Visual culture is woven throughout physical culture. Scholarship that integrates sporting themes with visual culture analysis is expanding. Researchers have explored how social identities of race, class, and gender are constructed in relation to representations of athletes in cinema (Baker 2006). O'Mahony (2006) has highlighted the role of sports culture and visual culture by examining how the Soviet Union propagated and presented stereotypes of dominance, i.e., powerful, youthful, athletic bodies. The statue of Smith and Carlos at San Jose State University, California has been reprinted in the *Journal of Sport History*, in which the authors advocate for a wider range of photographs, film, and monuments to help understand cultural memory "outside the written word" (Phillips et al. 2007, p. 287). Photography technology and what Huggins calls our "culture of looking" (Huggins 2008, p. 316) in the visual turn is expanded upon and applied in the following section.

## 3. Method and Procedure

The present study employs a contextual analysis of one photograph from 1964 of iconic Nation of Islam minister Malcolm X and luminary/famous professor/scholar–activist Dr. Harry Edwards. The current paper focuses on theory, textual analysis, and the deconstruction of the photo's content, imagery, and broader meanings. A contextual analysis accounts for the ways in which visual materials function within broad social ecologies:

> We need to expand the concept of context further still—to identify events as well as human agencies and convergences acting on and through the work. Context can help us to find our way through visual material where meaning can be slippery or multiple, but it

needs to be set in broad movements and theoretical structures, whether those of Marxist materialism, ideology, cultural hegemony, or feminism. (Huggins 2008, p. 319)

Thus, the historical document presented below is an analysis of race, sport, and activism within the visual turn. The artifact is a black and white photo of Dr. Harry Edwards standing next to Malcolm X in 1964 in Harlem, New York (see Figure 1).

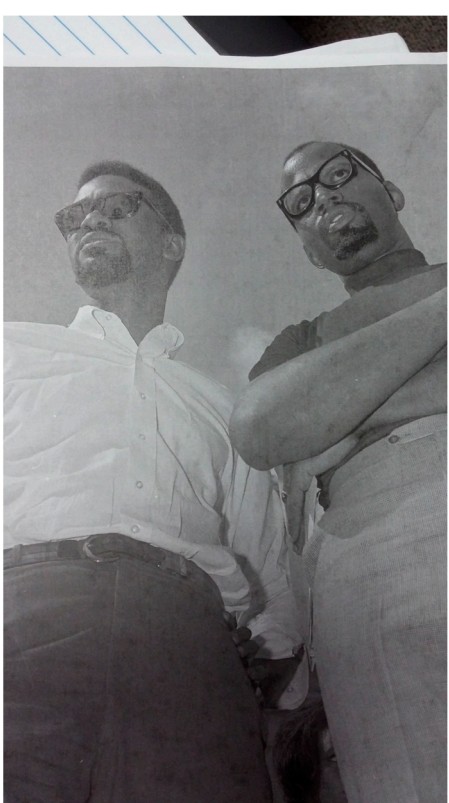

**Figure 1.** Malcolm X and Dr. Harry Edwards, 1964. (With permission from Dr. Harry Edwards.).

### 3.1. Photography and Visual Culture

The gaze is defined by theorists as a dual-layered perspective that captures both the ways viewers look at images of people in the visual medium and the gaze of the human subject within the visual text (Manlove 2007; Mulvey 1975). Gaze theory suggests that ways of seeing can be captured, evaluated, and explained. In the context of the 1960s, ways of seeing Black men were disempowering and disenfranchising. This is relevant to the current image because gaze theory accounts for the power structures in social institutions and conditions that created the image (Foucault 1991). In the photo, both subjects are framed as powerful beings. Upon first glance, the angle of the camera is most apparent because it accentuates the height of both subjects. Each man is already tall, with Malcolm X standing at six foot three inches and Edwards at six foot eight inches. Neither of them is stooped; both stand erect and attentive to what lies ahead. The inferior angle of the camera positions the towering men in a larger-than-life superiority. Giants in the frame, the pair of men metaphorically and physically embody the crusader spirit that drove them to advocate for Black power.

Traveling up the photo halfway, Edwards' long legs rest right below his crossed arms. Typically, crossed arms are interpreted as a defiant or protective stance. Taken together with his furrowed brow and fixed stare into the distance, his arms can also be read as a posture of deep concentration. Perhaps drawing upon his football-playing days, such focus in the moment is reflective of when athletes are "in the zone." Furthermore, his crossed arms are symbolic of the armed self-defense, self-determination, and Black pride espoused by the NOI. Similar to Edwards' stance, Malcolm X's legs occupy the bottom

third of the photo and give way to arms held akimbo. With his left hand on his hip and his gaze into the distance (right arm out of frame), his posture is open, ready, and alert. In accordance with the messages of Black nationalism and unity, each man is well-groomed and spotless. The creases in Edwards' pants match the crispness of Malcolm X's shirt collar. In relation to each other, the two men have overlapping bodies that cannot be more than a foot apart. The physical closeness of their bodies symbolizes the message of unity of the Black Panther Party. The verticality of their postures reflects the power, unity, and assertive vigilance that Smith and Carlos would build on with their extended arm. Using a contextual analysis, the content potentially suggests a constructed narrative for the image beyond the straightforward account of two determined Black men.

In addition to the content, the form of the photo provides additional layers of analysis. The grainy texture of the black and white film refers to the documentary-style news reporting of the time, lending it authenticity (O'Mahony 2018). The shallow depth of field ensures that the viewer's attention is solely focused on the Black men alone, which illustrates the dominating power that they impose. The tilt of the camera elaborates on the drama, effort, and gravity of performing Black masculinity in a country that kills Black men for existing. Such an approach to analysis situates the potential importance of photographs as sources of new insights into "how knowledge about the past is produced" in materiality and the rich texture of sport history (Osmond 2010, p. 119).

*3.2. Analysis and Broader Context of Images in Sport and Race*

The intersection of race and religion challenges researchers to synthesize critical scholarship on community ideologies from both domains. One potential approach to this synthesis is accomplished by turning to the tradition of sport. In recent religious studies scholarship, tradition is operationalized as "a set of practices, including styles of reasoning, that grows out of a shared history, has shared values implicit within it, and is supported by institutions" (Lloyd 2013, p. 82). Sport in the Black community has grown out of a shared history of discrimination, the performance of masculinity, and freedom of physical expression. The shared values of racial uplift brought the two mentors in the photograph together. This visual analysis of the Black bodies who support the athletes, as opposed to the athletes themselves, foregrounds the mentors who are usually behind the scenes of world-famous Olympians. This notion of the Black mentor contradicts the media portrayal of Black masculinity for several reasons. They are both standing tall, commanding the attention of the viewer with the powerful postures they exude. Their construction of manhood embodies the guardian positionality of both the Nation of Islam and the Black Panther Party. Rather than the stereotypical images that perpetuate powerlessness and subservience, Dr. Edwards and Malcolm X stand tall like giants.

Beyond the physical act of looking, the sporting gaze is a way of seeing that concerns human subjectivity (Huggins 2008). Adopted from media studies and art history, the sporting gaze allows an analysis of contextual factors that influence athletic events. Thus, while the photo evidence does not contain sporting events, it does frame two Black men who directly influenced the worldwide images that made the 1968 revolt of the Black athlete famous during the Olympic Games. The act of looking at Malcolm X and Edwards provides insight into the collaboration that resulted in Tommie Smith and John Carlos posturing themselves in a Black Power salute on the podium to receive their medals. The gaze is not immune from race, gender, sex, and class lenses: "We view art through stereotypes, and inversions of our self-image, thus creating the 'other,' treated perhaps with hostility or contempt, fear or condescension" (Huggins 2008, p. 321). The framing of Malcolm X and Edwards is a direct challenge to Black male inferiority at a time when Black men were portrayed as apes in visual media. Within the text itself, Malcolm X and Edwards gaze off into the distance relative to the camera angle. Both men have furrowed brows as they stare out, leaving the viewer to speculate about the future that these men envisioned together as they made waves of activism.

The contextual analysis is framed in the broader sociocultural context of the media paradox that reduces and situates Black masculinity in an illusion between those who deify their athletic prowess and those who despise them. On one hand, the glamour and spectacle of the celebrity Black male athlete displays a performance of wealth of an exceptional minority. In contrast, American society is also "subjected to the real-time devastation, slaughter, and body count of a steady stream of faceless black males on the 6 and 11 o'clock news" (Guerrero 1995, p. 396). It is important to learn how to discern between what is authentic and what is an identity performance instigated by a White gaze. In his iconic text, *The Soul of Black Folk,* W. E. B. Du Bois wrote of the psychological effect of the White gaze, where Black folks are in " . . . a world which yields him no true self-consciousness, but only lets him see himself through the revelation of the other world" (Du Bois 1997, p. 5). The othering of the White gaze in sport renders Black bodies hypervisible and scrutinized (Rehal 2020) and produces false racial inferiority used to justify Western dominance (Oh 2019). The White gaze circulates Black male images as a paradox rather than a spectrum of possibilities.

One contextual reading of the performance of Black masculinity in the photo is the element of the cool pose. In response to racial inequity, this defense mechanism is defined as, "a ritualized form of masculinity that entails behaviors, scripts, physical posturing, impression management, and carefully crafted performances that deliver a single, critical message: pride, strength, and control" (Majors and Billson 1992, p. 4). A negative dilemma of the cool mask is alienation from both the dominant culture and their own Black communities. As such, Majors and Billson theorize that an overemphasis on Black masculinity leads to a maladaptive model of male expression essential to the cultural cooperation and survival of the Black community. Unidimensional representations of cool Black manhood, such as the buck or gangster, are problematic because they do not provide strategies for dismantling the obstacles of Black masculinity.

Different gazes in visual culture can counter the cool pose as a maladaptive defense mechanism. A Black gaze is a more progressive construction of Black masculinity (Boylorn 2017). Such a gaze reveals the options of manhood outside of the White imagination, " . . . within Blackness, most notably constructions of black masculinity produced by the middle-class wing of the civil rights movement . . . " (Gray 1995, p. 403). This type of gaze allows room for images of Black men as "noble warriors in the case of Afrocentric nationalists and Fruit of Islam" (Gray 1995, p. 402). The image of Malcolm X and Edwards positions Black masculinity in opposition to the visual representations of Black masculinity as irresponsible, hypersexualized, and uncivilized. Their gaze is one of scheming, plotting, and looking to the future. This image marks a racial and cultural boundary of a counter-hegemonic representation of Black masculinity: "it is the ideal of the strong uncompromising black man, the new black man, the authentic black man, which anchors the oppositional (and within a nationalist discourse) affirmative representation of black masculinity" (Gray 1995, p. 404). Conchas et al. (2012) documented how young Black male students interpreted media portrayals of Black men as less capable and potentially dangerous. A Black gaze interprets the two men in the image as potentially dangerous because they are intellectually capable of creating waves of activism.

## 4. Discussion

Visual analysis of primary documents can illuminate nuanced aspects of sociopolitical movements and sociocultural practices. Through the visual turn, findings reveal the historical dynamics of Black manhood and the paradoxical representations of toxic masculinity that are widely circulated. This is in response to the call for new, better readings that move past binary images in the media (McCune 2012). The current paper makes a few contributions to the emerging literature. First, while much scholarly attention has been paid to the representation and iconic photographic moments of the 1968 Olympics, which was included in Life Magazine's (2003) anthology of one hundred photographs that changed

the world, little attention, if any, has been paid to other stakeholders connected to the Black athlete revolt in the 1960s.

The current paper analyzed the mentor of Smith and Carlos, Dr. Harry Edwards, and the historical iconic figure Malcolm X to fill a gap in Black male representation and create resistive spaces for defining progressive Black masculinity. Such spaces include a capacity to be fluid, feminist, vulnerable, affectionate, and resistant to patriarchy to reimagine gender roles (Neal 2006). Contemporary images allow us to dispute the existing confines of commercial visual culture and engage in the negotiations, challenges, and constant shifting of Black masculinity. In the literature that departs from deficit frameworks, findings indicate that Black men operationalize manhood as rooted in interpersonal and intrapersonal relationships, especially familial commitment (Givens et al. 2016; Goodwill et al. 2019; Hammond and Mattis 2005; Young 2004). The image of Malcolm X and Edwards transcends unidimensional portrayals of Black men with the stoic intellectualism usually reserved for the hegemonic culture.

Another contribution that the contextual analysis of the photo artifact offers is a diverse and complex way of envisioning and reimagining Black masculinity and collective identities of Blackness. O'Mahony (2018) called for a more critical approach to visual materials documented within sport history. Rather than employing a perspective that ignores the capacity for Black men to use their voices to define themselves (Hunter and Davis 1994), the present data analysis provides agency to reimagine Black identities as more inclusive of spirituality and humanism. It was their overlapping interests in Black religious practices that brought Malcolm X and Dr. Edwards into the same photograph. When asked to comment on the photo of himself and Malcolm X, Dr. Edwards noted the synergy represented in the artifact: "Religion and sports, role models, social class, etc., it is all connected" (Edwards, personal communication, March 2022).

The community of care and the role of a spiritual advisor in community-based organizations are integral to positive identity development. Thus, the pedagogy of care in Black religious communities is part of the sociocultural context of their photo. Their religious interest is significant because in White spaces, Christianity is often used to disguise racism (Allen 2019). Givens et al. (2016) found that establishing a caring community facilitates modeling manhood for Black males and supports the resistance of problematic mainstream representations in society and school. Through their in-depth analysis of qualitative interviews with young adult Black men, participants revealed how their attitudes and beliefs about Black masculinity were impacted by social movement figures and high-profile professional athletes. Interestingly, participants were especially influenced by athletes who faced public backlash from being in the spotlight of fame. This content helped ignite the role of a spiritual advisor to model emotional management (Givens et al. 2016), which is certainly what Smith and Carlos needed in the aftermath of 1968. Budding research continues to highlight how Black male youth construct healthy masculinity within the context of a faith-based positive youth development sport program (Newman et al. 2022). Community-based youth organizations should continue to provide agency to reimagine Black identities as progressive beyond pervasive biases and binaries, as Black students in high school develop strong proactive racial identities that symbiotically feed into academic success (Carter 2010).

An alternative reading of the visual analysis could signify an interpretive framework that does not read the image as a political act. Often, the evaluation of Black visual culture is fused with acts of resistance. Rather than overidentify with the social audience of a political aim, the theory of Black quiet approaches Carlos and Smith's iconic moment of resistance without the biased framework of resistance (Quashie 2009). Using this new method, Quashie draws attention to the deep intimacy, human vulnerability, and inwardness that Smith and Carlos posture themselves in. This visual cultural studies approach prioritizes the spiritual component of the intentional public demonstration in 1968. The surrendering bow of the head as if in prayer was an instance of visual quiet that reframes the meaning of Black hypervisibility, "wherein Blackness just *is* (Kaimana 2016, p. 147). Black quiet does

not imply nonaction. Rather, its expressivity has gained momentum in more recent protests by Black athletes.

The media has played a significant role in the production and circulation of images that reinforce dominant racist ideologies. Sports video gaming, for example, is a multibillion-dollar industry that commodifies the Black male body:

> "Eight out of ten black male video game characters are sports competitors; black males, thus, only find visibility in sports games. Just in larger society, the video game industry confines (and controls through image and ideology) black men to the virtual sports world, limiting the range and depth of imagery." (Leonard 2004, p. 1)

Even fantasy images of Black male athletes are informed by the White gaze onto Black bodies, allowing White players to indulge in "high-tech blackface" (Marriott 1999, p. 1). The virtual play through the White gaze profits from creating muscle-bound Black characters on games such as *NFL Street*.

## 5. Conclusions

Effective mentors know how to operate behind the scenes. Absent from the sport literature is visual evidence of the masterminds who used the media to broadcast a worldwide message of racial pride. As the revolt of the Black athlete in 1968 has demonstrated, the combination of race, religion, and sport links manhood to the collective human condition. There is additional evidence that documents the ripple effect of progressive Black masculinity. One of the authors has evolved the Black Power salute of the 1968 revolt to create an image of Black intellectualism and excellent character: the scholar-baller (see Figure 2).

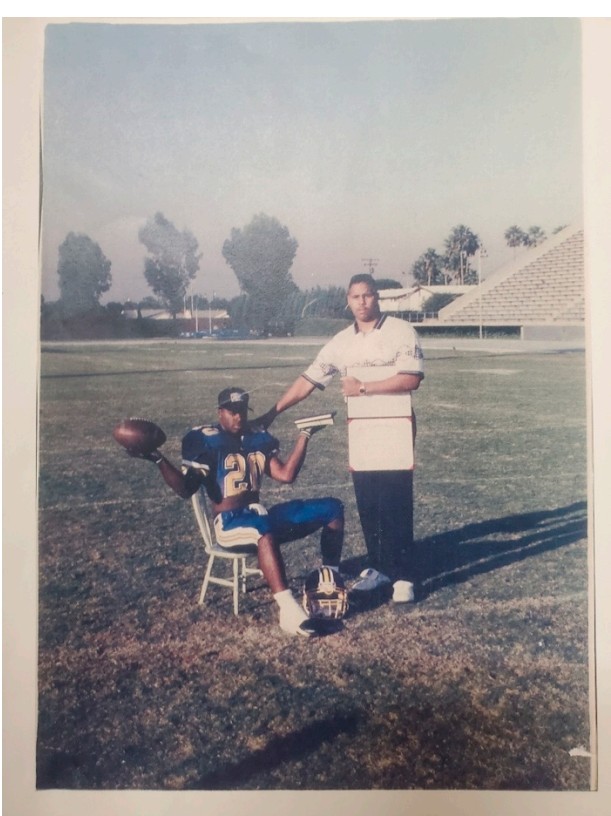

**Figure 2.** C. Keith Harrison and Otis C. Harrison Jr. at Picture Day, Cerritos College, 1992.

Coined in 1992, the scholar-baller is "an educated individual who also participates in sport, art, music, or any other extracurricular activity" (Harrison et al. 2010, p. 860). Now a fully developed program, the culturally inclusive pedagogy of the Scholar-Baller

model infuses popular culture into the academic framework to arouse the intellectual growth of student-athletes. As a result, student-athletes are empowered to affirm both their academic and athletic identities in order to reduce attrition rates. This model is an example of intentional curriculum design that accelerates the academic and social integration of Black male student-athletes toward successful matriculation throughout college (Dexter et al. 2021). It should be noted that the author who pioneered and implemented the Scholar-Baller model was directly mentored by Dr. Harry Edwards. And so, the waves of effective mentorship continue to advance new roles for Black masculinity as it is manifested in sports.

Future research could employ photo elicitation, such as the image above, to explore the connections between religion, sport, and race. The application of photo elicitation is a promising area of evaluation research in the visual sociology of religion (Williams and Whitehouse 2015). For example, a researcher could use several images at the intersection of race and religion in an interview with participants to discuss their experiences in sport programs, invite reflections on effectiveness, and solicit suggestions for improvement of such programs. Alternatively, photos taken at the beginning, midpoint, and end of a sport program for Black Christian athletes may be helpful in isolating moments of continuity and change (Hurworth 2004).

This new wave of intellectual and political depth in 1992 has continued to ripple into Colin Kaepernick's activism with one thread of commonality: Dr. Harry Edwards' mentorship. From standing tall in 1968 to sitting as a scholar in 1992 to kneeling in faithful supplication in 2016, Edwards and Malcolm X made waves of activism that are still in effect. If we want to precipitate Black Lives Matter movement success, athletes, spiritual leaders, and educators should be involved more than ever.

**Author Contributions:** Conceptualization, W.G. and C.K.H.; methodology, W.G.; validation, W.G. and C.K.H.; formal analysis, W.G.; investigation, C.K.H.; resources, C.K.H.; data curation, C.K.H.; writing—original draft preparation, W.G.; writing—review and editing, W.G.; visualization C.K.H.; supervision C.K.H.; project administration, W.G. All authors have read and agreed to the published version of the manuscript.

**Funding:** This research received no external funding.

**Institutional Review Board Statement:** Not applicable.

**Informed Consent Statement:** Not applicable.

**Data Availability Statement:** Not applicable.

**Conflicts of Interest:** The authors declare no conflict of interest.

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
