# Peer review of "Giants in the Frame: A 1964 Photo Analysis of How Malcolm X and Dr. Harry Edwards Connected Race, Religion, and Sport"

_religions, doi:10.3390/rel14050580_

Round 1
Reviewer 1 Report
This essay uses a unique frame, the analysis of a photograph, to illuminate the role played by Harry Edwards and Malcolm X in advancing a new appreciation for and understanding of Black masculinity as it is manifested in sports. The authors show a great familiarity with the literature relevant to Black masculinity and the protests of the 1960s and did an excellent job explaining the role sports played for Black nationalism in that era. The analysis of the photograph was a brilliant explanation of the power of the image to reframe black masculinity. There are two major (and related) areas the authors need to address to make this essay work for inclusion in a volume about the Collision of Race, Religion and Sports however.
In terms of the photograph itself, it shows Dr. Edwards and Malcolm standing together, but does not tell the story of the connection between them. Where are they? What brought them together? What do we know about the history and the origin of the photograph? The photograph and its context should be more central to the essay.
The reader also needs to know more about both Dr. Edwards and Malcolm to make it matter that this photograph is of them and not any two powerful Black men of that era. The little background given about their connection to religion and to each other (lines 125-128) is not developed and flawed in its understanding of religion. How did Dr. Edwards’ MA thesis about Black Muslim religion bring him into “Malcolm’s orbit”? What does that phrase mean? Did they meet? Exchange letters or ideas? The next sentence, “The topic of Edward’s thesis on the Black Church revealed (in part) the affinity of Black sports heroes by Black Americans during that time” (127-128) is troubling. Are the authors conflating the Black Church and Black Muslim religion? One could assume so since the rest of the discussion about religion is only about the Christianity and the Black Church and not about Black Muslim religion. Calling Muslim worship spaces churches and not mosques obscures the way NOI distinguished itself from the Black Church, which they identified as a major barrier to Black advancement. The essay shows no understanding of the deep religious commitments of Malcom X to NOI or later to Islam, or how those commitments might have played a significant role in the dialogue between him and Dr. Edwards or the role of sports.
I like the point made later in the essay to explain Dr. Edwards’ mentorship of Tommie Smith and John Carlos as spiritual and part of a Christian ethos of care, but I see no evidence for the conclusion that it was “their overlapping interests in Black spirituality that brought Malcolm X and Dr. Edwards into relationship” (501). The essay needs to reveal more about the connection between these two men to support that assertion. I would imagine that a conversation with Dr. Edwards might reveal some interesting dimensions to this relationship that could be incorporated into the essay.
To sum up: the only insight related to religion in this essay is about framing Dr. Edwards as a spiritual mentor to Smith and Carlos (and I might presume, also to Kaepernick). The essay makes a significant contribution to our understanding of race and sport but does not adequately deal with how religion figures into the analysis. This is not a flaw in the research; the essay includes a strong bibliography and makes key references to recent works on race, Christianity, and sport. But an explanation of the role of Islam in Malcolm X’s relationship to Black masculinity, a deeper explanation of his (and Islam’s) influence on sports figures like Ali and Abdul-Jabbar, and his conflictual relationship to the Black Church need to be incorporated. Finally, a stronger narrative about the relationship between Dr. Edwards and Malcolm that resulted in that photograph as well as their different views on the role of religion for black masculinity and sport would be necessary to make this essay as powerful as I suspect it could be for this issue.
Reviewer 2 Report
Overall, this is a wonderful paper. It well structured and has a strong argument. I am grateful I was selected to review it and look forward to reading more scholarship of a similar focus. There are hundreds, if not thousands, of images which could be analyzed in this same way, and I would encourage the author to do more of this in future writing.
Three areas of improvement, with the first two being the most pertinent:
1) In the section entitled, "Analysis and Broader Context of Images in Sport and Race", the author analyzes the image from sporting and racial viewpoints, but does not include anything about spirituality or religion. This is briefly alluded to in the next section, "Discussion", but should also be addressed in the analysis section. If the paper is on the intersection of sport, religion, and race as is depicted in visual images, all three areas should be addressed in the analysis of the image under review.
2) The conclusion discusses the need for mentorship and people to work behind the scenes to reshape the image of black, male athletes in the USA. The sentiment and message are beneficial, but it feels disconnected from the rest of the paper. Could a more traditional conclusion paragraph be added to bridge between this conclusion and the majority of the text?
Another idea would be to add the concept of mentorship/behind the scenes work to the analysis section. The image of Malcom X and Dr. Edwards used depicts them behind the scenes, supporting athletes. This notion of the black mentor goes against the media portrayal of black masculinity described in the text for multiple reasons. For example. it focuses on the black body supporting the athlete, as opposed to being it. Addressing this in the analysis section strengthens the conclusion.
3) In the section entitled "Image Representations of Black Men in American Culture" it may be beneficial to add a few lines on how the black male is typically represented in images within American culture. As this article focuses on black male athletes this feels particularly important to address in this section.
A few sentences or paragraphs would go a long way in tightening up these two areas. Overall, a well done article.

Reviewer 3 Report
The underlying thesis is weak as are the arguments and the way in which they try to support the objectives mentioned in the introduction. In order to propose a scientific article, solid arguments supported by data and related literature are needed. The article claims to link religion and the delicate racial issue and sport: the first two topics are very complex and require an adequate introduction. The discourse of sport must be included in a more capillary and articulated way. References and underlying arguments must be solid.
Round 2
Reviewer 1 Report
Thank you for your thoughtful response to my reviewer's report. Your additions and clarifications met all my concerns, and I will look forward to seeing this "in print."
Author Response
Thank you for your thoughtful response to my reviewer's report. Your additions and clarifications met all my concerns, and I will look forward to seeing this "in print."
Reviewer 3 Report
I understand the effort made for the in-depth study. This certainly improved the article. But, as already mentioned, the basic assumption is weak. If the intention is to support the thesis, then it is strongly recommended to first build a solid background presenting the thesis, with references to texts from the scientific literature and then introducing the author's central topic of interest.
Round 3
Reviewer 3 Report
Twice I have already strongly recommended giving a solid structure to the article. Even in this version, I see that this point has not been developed. I have also already stressed that the basic hypothesis thus presented is not solid and needs to be introduced by a broader and more scientific presentation. This point has also not been developed. These two aspects are necessary for publication in a scientific journal.